POINT OF VIEW

# The challenges faced by living stock collections in the USA

**Abstract** Many discoveries in the life sciences have been made using material from living stock collections. These collections provide a uniform and stable supply of living organisms and related materials that enhance the reproducibility of research and minimize the need for repetitive calibration. While collections differ in many ways, they all require expertise in maintaining living organisms and good logistical systems for keeping track of stocks and fulfilling requests for specimens. Here, we review some of the contributions made by living stock collections to research across all branches of the tree of life, and outline the challenges they face.

KEVIN MCCLUSKEY, KYRIA BOUNDY-MILLS, GREG DYE, ERIN EHMKE, GREGG F GUNNELL, HIPPOKRATIS KIARIS, MAXI POLIHRONAKIS RICHMOND, ANNE D YODER, DANIEL R ZEIGLER, SARAH ZEHR AND ERICH GROTEWOLD[*]

**\*For correspondence:**
Grotewold.1@osu.edu

## Introduction

The goals of living stock collections are to preserve the genetic diversity of target organisms, to maintain research materials, and make these resources available to researchers around the world. Living stock collections are distinct from other bio-repositories, such as natural history museums (*Rocha et al., 2014*) and biobanks (*Baker, 2012*), because the resources they contain are generally capable of being multiplied and propagated. This creates unique challenges for long-term sustainability.

The collections are typically housed within stock centers, seed banks, vivaria and botanical gardens, which are usually based at a university or other research institution. Collections make their resources available in a number of ways: these include distributing resources to qualified researchers, providing access to materials at the collection for specific experiments, and the sharing of detailed historical information regarding each organism or strain.

Living collections have been identified as the foundation of the emerging bioeconomy (*OECD, 2001*) and they significantly increase the impact of shared research materials (*Furman and Stern, 2011*). By allowing access to identical strains, cultivars and cell lines, the collections allow published research to be directly reproduced. This is of special value because – along with addressing concerns about the reproducibility of scientific data – it also makes individual organisms, clones, populations or tools that have been used successfully in research studies available to other investigators, bypassing the need for repeated optimization studies.

Living collections are funded by a number of mechanisms. In the United States, for example, the Department of Agriculture Agricultural Research Service (USDA-ARS) supports several centers that conserve and distribute germplasm of agricultural importance. Similarly, the National Institutes of Health (NIH) maintains diverse collections of animal models of human disease such as rodents, swine, axoltls and primates. Finally, the National Science Foundation (NSF) has supported diverse living genetic and biodiversity collections for over 50 years through a competitive program now called Collections in Support of Biological Research (CSBR).

The global research and development community values living collections as demonstrated by recent progress in the development of networks to create a global microbial research commons (*Dedeurwaerdere, 2010*; *Uhlir, 2011*). These efforts are bearing fruit in the number of growing networks, consortia and even

international treaties on access to genetic resources. The ratification and activation of the Nagoya Protocol on Access and Benefit Sharing in 2014 (*Dedeurwaerdere et al., 2012*), and of the International Treaty on Plant Genetic Resources for Food and Agriculture in 2004 (*Mekouar, 2002*), has required that research and development consider the place of origin in sourcing research materials. Living collections are key partners in ensuring that materials are ethically and legally procured (*Boundy-Mills et al., 2016*).

Our focus here is on open living research collections in the USA that are funded by a combination of competitive grants and community user fees (*Table 1*). Many of these collections were assembled over multiple decades and would be difficult or impossible to replace. We emphasize that these resource centers are essential for the long-term maintenance of key living resources for research and scientific replication and as such they are highly vulnerable to policy and funding changes. This creates dangerous uncertainty for the communities affected.

If the centers that harbor these collections cease to exist, or even if their operations must be reduced below a certain critical threshold, the negative consequences to the scientific community are unavoidable. For example, without stock centers there is an increased risk of researchers using inauthentic materials (such as contaminated or improperly identified stocks), research communities may become more exclusive, and it may cost more to generate key strains, clones, lines or varieties. Ultimately, this makes it harder for researchers to reproduce key results (*Sheppard, 2013*).

## Impact of living collections on research

Living collections impact research at many different levels. At the most basic level, they provide the biological resources for fundamental studies. In one high profile example, the repeat sequences now called CRISPR were first observed in a phosphatase mutant strain of *Escherichia coli* (*Ishino et al., 1987*) generated in a mutant screen that used strains from the *E. coli* Genetic Stock Center, which is supported by the NSF (*Nakata et al., 1978*). Similarly, the first experiments to demonstrate the polymerase chain reaction were conducted using an enzyme isolated from a thermophilic bacterium that had been deposited into the American Type Culture Collection almost twenty years earlier

(*Mullis et al., 1986*). The *Penicillium* strain that has been used for large-scale antibiotic production since the mid-1940's (supplanting the original Fleming strain) was isolated and shared through the USDA NRRL collection (*Raper et al., 1944*), therein launching the modern era of antibiotics.

Living collections are important for national security and have been used in many situations including the 2001 Anthrax attacks (*Kurtzman, 2011*) as well as to identify the source of infection in an outbreak of the eye disease ocular keratitis (*Short et al., 2011*). Similarly, through identifying pathogenic organisms associated with agriculture, and breeding for resistance to emerging plant and animal pathogens, living collections are foundational for food security. And, because they are central resources for student projects and often repositories of protocols and technical expertise, living collections help train new generations of students to be researchers and scientists.

Living collections also provide an invaluable resource to help solve the irreproducibility problem that is plaguing the scientific literature (*Sheppard, 2013*). For example, stock centers have been identified as key players in ensuring the integrity and identity of natural isolates or ecotypes (*Anastasio et al., 2011*) and in providing quality controlled lines for biomedical research (*Stacey, 2000*). Living collections also help to ensure that plant genetic resources are preserved and accessed ethically (*McCouch et al., 2013*), and the Convention on Biological Diversity has identified them as the appropriate means for us to preserve and benefit from microbial biodiversity.

## Living collections capture an important, yet minor fraction of extant biodiversity

Historically, living collections have generally focused on organisms that serve research communities of significant sizes, often corresponding to model systems that have been broadly embraced by the community (*e.g., E. coli, Neurospora crassa* and *Arabidopsis thaliana*). In some cases, such as certain fruit fly species in the genus *Drosophila*, the stock center is the only source of these stocks as they can no longer be collected in the wild.

The ability to culture microorganisms previously believed to be 'unculturable' [see for example (*Browne et al., 2016*)], combined with using genomics information to validate

**Table 1.** A selection of public living research collections in the USA.

| Collection name | Acronym | Holdings | Host | Support |
|---|---|---|---|---|
| Microbial collections | | | | |
| American Type Culture Collection | ATCC | 18,000 bacterial and 7,600 fungal type strains | ATCC | Users, government contracts |
| BEI Resources | BEI | 13,000 strains and reagents for emerging pathogen research | ATCC | NIAID |
| Fungal Genetics Stock Center | FGSC | 25,000 filamentous fungi including mutants, genetic testers, wild strains, plasmids and mutant sets | Kansas State University | NSF (1961–2014), KSU, user fees |
| Phaff Yeast Culture Collection | UCDFST | 7,500 wild-type yeast | University of California, Davis | UC, NSF, user fees |
| E. coli Genetic Stock Center | CGSC | 8,000 mutant and wild K12 E. coli | Yale University | NSF, user fees |
| Bacillus Genetic Stock Center | BGSC | 2,600 mutant and wild Bacillus subtilis | The Ohio State University | NSF, user fees |
| International Culture Collection of (Vesicular) Arbuscular Mycorrhizal Fungi | INVAM | 1,112 vesicular arbuscular mycorrhizal fungi | West Virginia University | NSF, user fees |
| World Phytophthora Collection | WPC | 10,000 wild oomycete fungi | University of California, Riverside | UCR |
| USDA ARS Culture Collection | NRRL | 95,000 agricultural and industrial fungi and bacteria | USDA National Center for Agricultural Utilization Research | USDA |
| USDA ARS Collection of Entomopathogenic Fungal Cultures | ARSEF | 13,000 fungal cultures | USDA Robert W. Holley Center Center | USDA |
| UTEX Culture Collection of Algae | UTEX | 3,000 freshwater algae | University of Texas, Austin | NSF, user fees |
| National Center for Marine Algae and Microbiota | NCMA | 2,800 algal cultures, viral and bacterial associates | Bigelow Laboratory for Ocean Sciences | NSF, user fees |
| The Chlamydomonas Resource Center | Chlamy | 4,000 mutant and wild type strains | University of Minnesota | NSF, user fees |
| Animal and cell line collections | | | | |
| Bloomington Drosophila Stock Center | BDSC | Over 50,000 Drosophila genetic stocks | Indiana University | NIH, user fees, HHMI |
| Duke Lemur Center | DLC | 250 living and 4,000 historic individual Strepsirrhine primates, with a biosample bank of >10,000 samples | Duke University | NSF, user fees |
| Drosophila Species Stock Center | DSSC | Flies | University of California San Diego | NSF, user fees |
| Jackson Laboratories | JAX | Mice | Jackson Labs | User fees |
| Peromyscus Genetic Stock Center | PGSC | At least 4 species and several coat color and behavioral mutants of deer mice | University of South Carolina | NSF, user fees |
| Plant collections and seed banks | | | | |
| Arabidopsis Biological Resource Center | ABRC | ~1 million Seeds and DNA Stocks | The Ohio State University | NSF, user fees |
| Maize Genetics Cooperation Stock Center | MGCSC | Over 100,000 maize variants | University of Illinois, Urbana/Champaign | USDA-ARS |
| National Plant Germplasm System | NPGS | 576,991 Plant accessions | Distributed around the US and backed up at the USDA NLGRP in Ft. Collins | USDA-ARS |

taxonomy and genetic properties, is increasing the number of new strains being deposited in living collections around the world (**Boundy-Mills et al., 2015**). Nevertheless, these collections continue to capture only a tiny fraction of the existing biodiversity, and this is likely to continue to be the case in the future.

For many purposes, the possibility to bank and distribute genomic DNA provides a simpler and less expensive alternative to storing the

whole organism, although microbial type and patent strains need to be preserved alive to satisfy taxonomic or treaty obligations. For larger organisms, such as plants, it is often necessary to develop specific practices for each species. For example, the procedures used to grow and preserve seeds of the model plant *A. thaliana* would not be suitable for maize or other cereal crops.

## Challenges to maintaining collection integrity

Collecting, preserving and making reference material available to the community requires living collections to maintain very strict quality control standards, regarding not only the viability of the stock, but also their identity and authenticity. Viability and purity checks have been an integral part of quality control at most stock centers for many years. Animal cell lines have suffered many problems with misidentification of stocks and contamination (*Hughes et al., 2007*), which is forcing the community to develop stringent standards for cell line authentication (*Almeida et al., 2016*). Stocks used to be identified on the basis of morphology and phenotype, which are affected by the way in which the organism's genes interact with the environment. However, the advent of easily accessible genomics tools has forced research communities and the corresponding living collections to shift to performing genotyping analyses, which are often significantly more time consuming, expensive and require specialized personnel.

More difficult to detect, but equally important, are instances of spontaneous mutations that arise as a consequence of key stocks that are used as references by the community being continuously replicated. This can lead to the stock changing so much that it is no longer a true reference. The plant community has been particularly vocal about this problem, developing a set of best practices to be implemented by researchers and stock centers to avoid it (*Bergelson et al., 2016*). Similarly, microbe collections reduce genetic drift by using techniques such as freeze drying and cryopreservation that preserve material in suspended animation and these practices are fundamental to published best practice guidelines (*Wiest et al., 2012*).

Materials in collections are usually deposited by independent researchers and may be exchanged between stock centers, which generates additional challenges in controlling the authenticity and equivalency of the stocks.

The microorganism community has partially solved this problem through the introduction of StrainInfo, a strain passport that captures all the exchange history of the stock, as well an overview of the strain in an uniform format (*Verslyppe et al., 2014*). To what extent a similar data integration and tracking system could be adopted by other communities is not clear, although a persistent uniform resource identifier would help deal with this issue.

## Research in the absence of living collections

While the impact of living collections has been amply demonstrated (*Furman and Stern, 2011*), not all research communities have the benefit of open collections. Although *Saccharomyces* has been a major research system with high impact – including results that have produced several Nobel prizes in recent years – stocks have been maintained without a formal centrally-managed yeast culture collection for many years.

The Yeast Genetic Stock Center collection, operated for several decades by R. Mortimer (*Mortimer and Johnston, 1986*), was donated to the American Type Culture Collection (ATCC) in 1998 and most gene deletion sets have been managed by commercial vendors. Most yeast strains and related materials were shared on a peer-to-peer, ad-hoc basis where individual investigators were free to limit distribution, creating a closed community that further complicates research reproducibility and open science. Moreover, the detailed breeding records maintained for decadal mammal collections give investigators assurance that the interpretation of data will not be inadvertently conflated by genetic relatedness.

Many research systems have dedicated living repositories and some enjoy significant economies of scale. Mice from the Jackson Laboratories, genetic stocks of *Drosophila melanogaster* from the Bloomington Stock Center, and diverse animal models of human disease are available from either commercial or publicly supported collections. Most microbe and biodiversity related resources do not have this scale, and as such are relegated to a more modest level of support, often driven by the initiative and efforts of the collection staff. While the research systems supported by these smaller biological resource centers have made tremendous impact over the decades, the collections face increasing challenges that threaten the ability of the

community to access diverse research systems effectively.

In the absence of open collections with their established quality management, researchers must resort to obtaining materials from colleagues or isolating similar organisms directly from nature, thereby running the risk that the materials are not identical across studies. The adage, "apples to apples" refers to direct comparisons, but to stretch the metaphor, it could be more accurately described as "Red Delicious apples to Red Delicious apples." Otherwise the risk is that comparisons are on the order of "Granny Smith apples to Crab apples," which, to the apple pie chef, is bound to yield disappointing results. Without this high degree of specificity, the ability to accurately produce comparable results across studies is diminished.

## The challenges ahead

The outcome of reduced support for living stock collections is disproportionately borne by small institutions, students and researchers in areas not tied to human health or other research systems with high economic impact (*Mccluskey, 2017*). By way of contrast, even modest support for living collections pays dividends to public, academic and scientific communities in many different ways.

Collections have both the capacity and the obligation to reflect developments in biological inquiry. Long-term support for collections can ensure that historical materials from one era are available to generate technological advances in the next generation, thereby enabling answers to research questions that were not envisioned when the materials were first collected, characterized and preserved. Open collections ensure the availability of such resources by implementing proven approaches to managing stocks – including modern resources such as plasmids and gene deletion mutants – and by developing novel culture methods to bring historically nonculturable organisms into the mainstream. With good quality and data management strategies they can also ensure that the associated information is standardized, easily retrievable and sharable with users, as is being done for the microbiology community (*Verslyppe et al., 2010*; *Wu et al., 2016*).

The NSF has funded living collections over many years and, as a direct consequence of their reporting requirements, the collections they support have longstanding quality management practices as well as robust data on the use of material and its impact, collection growth and sustainability. Accordingly, NSF-supported collections have long histories of implementing best practices (*Wiest et al., 2012*) that ensure access to high quality resources. USA federal support requires that collections maintain detailed records, a formal community advisory board evaluates each collection's holdings and practices, and that the collections share resources without regard to personal preference, historical relationship, or even institutional affiliation. Living public collections "level the playing field" and allow equal access to valuable, well-documented materials. Coincidently, funding agencies also benefit from supporting living collections given that the collections are natural partners in material management plans.

With the input of formal advisory boards, living stock collections speak on behalf of their research communities and are therefore placed in the uniquely awkward position of having to advocate for their own continuance. Shared metrics, such as a pseudo *h*-index that records the number of citations to publications generated via use of the collection, are useful in communicating the value and impact of living collections. Several living collections have pseudo *h*-indices on the scale of 60–125. Other collections have too many citations to use available *h*-index calculations. For example, the ATCC is cited over 600,000 times in the Google Scholar database, and the USDA Agricultural Research Service NRRL culture collection has documented over 49,000 citations that directly work with strains in the collection.

These measures are imperfect and a quantitative mechanism to document how resources in living collections are used might be a powerful mechanism for further establishing the value of federal investment in these collections. A global identifier for research resources such as strains, cultivated varieties, cell lines and animals would be a valuable first step in this process (*Wu et al., 2016*). In addition, adopting policies similar to those employed recently to authenticate cultured cell lines could also be applied.

While the International Code of Nomenclature of Algae, Fungi and Plants (*McNeill et al., 2012*) requirement that new type strains be deposited in at least three public collections in at least two countries is a good model, the number of modified strains used in public research would overtax the capacity of present collections. This notwithstanding, the authentication of specimens' identity through available records of living collections could be considered

sufficient to the extent that the collection follows best practices for living collections and bio-banks. This also argues that living collections seek and obtain external certifications, such as those available through the International Standards Organization (ISO) or the Good Laboratory Practice as described by the Organization for Economic Cooperation and Development.

Another complicating factor that living collections face is the non-uniformity in resource ownership, which has several facets. First, different agencies have different ownership standards. For example, USDA collections are all owned by the USDA, and most NIH collections are owned by the NIH. Conversely, collections that receive NSF support are owned by their host institutions, or are maintained and distributed on behalf of the donor. While many collections consider that their resources are in the public domain, they are more accurately held in trust for the public (*Uhlir, 2011*).

Second, most living collections in the USA have been assembled over many years, often several decades, and little attention has been given to formal transfer of intellectual property rights. Modern collections require both material accession agreements and, for subsequent distributions, material transfer agreements (MTAs). These agreements typically limit both rights and liabilities and can assume a variety of levels of rigor, ranging from implied, to "click-through", to formal. For example, the Addgene plasmid collection has been assembled with intellectual property management at the forefront, simplifying subsequent distribution of resources (*Kamens, 2015*; *Kamens, 2014*). European microbe collections, united by the European Culture Collection Organization, embrace the TRUST code of conduct – which addresses both MTA issues as well as compliance with the Nagoya Protocol on Access and Benefit Sharing.

USA culture collections addressed the question of how to ensure compliance at an NSF-sponsored meeting in February. This meeting was open to collaborators from every domain of life, and included participants from natural history collections, as well as living research and biodiversity collections (http://www.usccn.org/Pages/USCCN_Nagoya_2017.aspx). As exemplified by the engagement at this meeting, staff at living collections are at the forefront of ensuring that ethical practices are followed in obtaining and distributing living resources. Importantly, the participants heard from the USA National Focal Point for the Nagoya Protocol that the USA does not restrict access to germplasm, although certain landowners or managers, such as the US National Park System, may have their own requirements for accessing genetic resources.

Additional presentations at the meeting emphasized that each party to the Nagoya Protocol is required to establish their own national legislation on Access and Benefit Sharing. Brazil and the EU have the most mature legislation, accessed via the Convention on Biological Diversity Access and Benefit Sharing Clearing-House (ABSCH, https://absch.cbd.int/). The highly divergent perspectives on what constitutes "access" emphasize that researchers should consult the ABSCH prior to using genetic resources (or information) with an origin outside their own country.

## Forward directions

Living collections benefit from public support to ensure that valuable resources for research in every area of biology are available to future generations of scientists (*Mccluskey, 2017*). While some medical and agricultural collections receive public funds, many public biodiversity and genetics collections do not. Without this external support, the collections managers have no alternative but to recover the costs of collection maintenance by raising user fees. While this simple approach is appealing, it creates a scenario where only well-funded laboratories can afford to obtain validated materials.

To ensure that the materials generated by today's research investment are available to future generations of scientists, living collections need basic financial support including salaries and subsidies on end-user fees. Living collections will benefit substantially if journal editors and granting agencies enact and enforce requirements that materials described in publications be available from public repositories, just as gene and genome sequences are required to be deposited in and distributed by public data repositories. Requiring capacity building beyond simply preserving the materials from the past will allow preservation and documentation of the large numbers of deposits generated by the requirement that living resources be available from public sources. Standing on the shoulders of giants is made easier by access to shared materials. The availability of authentic and diverse materials from published research empowers all investigators, regardless of their career stage or funding status.

## Acknowledgements
Many of the ideas presented here were formalized during an NSF-supported workshop on the power of biological infrastructure to advance knowledge (DBI-1642534 to ADY). This is DLC publication number 1343. This is contribution 17-233-J from the Kansas Agricultural Experiment Station.

**Kevin McCluskey** is in the Department of Plant Pathology, Fungal Genetics Stock Center, Kansas State University, Manhattan, United States

**Kyria Boundy-Mills** is in the Phaff Yeast Culture Collection, Food Science and Technology, University of California, Davis, Davis, United States

**Greg Dye** is in the Duke Lemur Center, Duke University, Durham, United States

**Erin Ehmke** is in the Duke Lemur Center, Duke University, Durham, United States

**Gregg F Gunnell** is in the Duke Lemur Center, Duke University, Durham, United States

**Hippokratis Kiaris** is in the Department of Drug Discovery and Biomedical Sciences and Peromyscus Genetic Stock Center, University of South Carolina, Columbia, United States

**Maxi Polihronakis Richmond** is in the Drosophila Species Stock Center, University of California, San Diego, San Diego, United States

**Anne D Yoder** is in the Duke Lemur Center, Duke University, Durham, United States

**Daniel R Zeigler** is in the Bacillus Genetics Stock Center, The Ohio State University, Columbus, United States

**Sarah Zehr** is in the Duke Lemur Center, Duke University, Durham, United States

**Erich Grotewold** is in the Arabidopsis Biological Resource Center, The Ohio State University, Columbus, United States

http://orcid.org/0000-0002-4720-7290

*Author contributions:* KM, KB-M, GD, EE, GFG, HK, MPR, ADY, SZ, EG, Writing—original draft, Writing—review and editing; DRZ, Writing—review and editing

*Competing interests:* The authors declare that no competing interests exist.

## Funding

| Funder | Grant reference number | Author |
|---|---|---|
| National Science Foundation | DBI-1534564 | Kevin McCluskey |
| National Science Foundation | DBI-1351502 | Maxi Polihronakis Richmond |
| National Science Foundation | DBI-1642534 | Anne D Yoder |
| National Science Foundation | DBI-1561691 | Anne D Yoder |
| National Science Foundation | DBI-1561210 | Erich Grotewold |
| National Science Foundation | DBI-1349230 | Hippokratis Kiaris |

The funders had no role in study design, data collection and interpretation, or the decision to submit the work for publication.

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
