## [Decision Letter]

Thank you for submitting your article "Public living collections and their role in biological research: Impact, challenges, and expectations" for consideration by *eLife*. Your article has been reviewed by two peer reviewers, and the evaluation has been overseen by a Reviewing Editor and Peter Rodgers as the Senior Editor. The following individual involved in review of your submission has agreed to reveal their identity: Paul De Vos (Reviewer #2).

The reviewers have discussed the reviews with one another and the Reviewing Editor has drafted this decision to help you prepare a revised submission.

Summary: The manuscript describes the important role of living stock collections in scientific research, underlining the major challenges that they face today.

Although the manuscript deals with a very important issue in biological research, the reviewers feel that there are some aspects that could be discussed in more detail to make the manuscript more valuable to the research community and improve the arguments for continued support of public living stock collections.

Essential revisions:

1) The importance and the necessity of financial support is mentioned too often in several different paragraphs. This gives the impression that the manuscript was written to promote the collections in order to ask for money (public money). Please cut, or at least revise, most of the sentences related to financial support and rewrite the manuscript accordingly.

2) The reviewers consider living collections to be indispensable as accessible references for taxonomic (hence identification) purposes. Please expand the discussion on this topic, including a reflection on the present situation about the coverage of the complete biodiversity.

More information from reviewer #2: "Preserving and making reference material available also imposes a very strict quality control (QC) on the viability and the authenticity of living stock cultures/collections that are maintained/preserved. One must realize that this QC is crucial and that a correct application of the implemented QC system is key for the operational value of the whole system. Indeed, living stocks are subjected to genetic drifting during their preservation period. If the cultures are used for taxonomic references, the taxonomic markers are usually stable enough for identification above the strain/specimen level. However, when markers are used for identification at or even below the strain level (isolate/specimen level) the reference strains may have undergone genetic changes that make them no longer suited as references for identification, or in those cases where the living stocks are used in certified test in e.g. pharmaceutical context."

3) Although the problem of integration of data is mentioned in the manuscript, it is not discussed deeply enough to underpin the lack of interest for the situation along with the lack of financial support.

More information from reviewer #2: "Researchers often exchange material in collections (mostly for free). In many cases, neither the donating nor the receiving party have the skills or the attitude to control the authenticity of the material that is exchanged resulting in the use of non-equivalent material for their research. This means that the results of cumulative research must be treated with a degree of criticism. Live stocks are also exchanged between collections and in this cases an important number of discrepancies of various nature are reported. If researchers plan to integrate data from different collections several controls need to be introduced as has been demonstrated by the various papers of the strainInfo team (Van Brabant B, Gray T, Verslyppe B, Kyrpides N, Dietrich K, Glockner FO, Cole J, Farris R, Schriml LM, De Vos P, De Baets B, Field D, Dawyndt P. 2008. Laying the foundation for a Genomic Rosetta Stone: creating information hubs through the use of consensus identifiers. IOMICS 12:123-127. Verslyppe, B., De Smet, W., De Baets, B., De Vos, P., Dawyndt, P. (2011). Make Histri: Reconstructing the exchange history of bacterial and archaeal type strains. Syst Appl Microbiol 34: 328-336. Verslyppe, B., De Smet, W., De Baets, B., De Vos, P., Dawyndt, P. (2014). StrainInfo introduces electronic passports for microorganisms. Syst. Appl. Microbiol. 37(1): 42-50 etc.)"

4) Include more in-depth discussion of some of the ethical issues surround living collections:

Often the research has been supported by public money and hence the data belong to the public domain. However, protective actions are seldom undertaken to prevent this data being used for private, commercialized applications. Although the international community tried to take care of this problem via treatments and conventions (CBD, Nagoya protocol). Unfortunately, these conventions are not recognized by all governments on the globe and are more like a burden.

Only about 10-15% of the biological material that has been the subject of research is available via public collections. It is clear that from a scientific viewpoint, this is unacceptable and should be of major concern.

---

## [Author Response]

*Essential revisions:*

*1) The importance and the necessity of financial support is mentioned too often in several different paragraphs. This gives the impression that the manuscript was written to promote the collections in order to ask for money (public money). Please cut, or at least revise, most of the sentences related to financial support and rewrite the manuscript accordingly.*

We eliminated most of the instances in which the manuscript promoted the collections as if asking for public funds, and modified the text accordingly. However, funding remains one of the challenges, hence it deserves to be mentioned.

*2) The reviewers consider living collections to be indispensable as accessible references for taxonomic (hence identification) purposes. Please expand the discussion on this topic, including a reflection on the present situation about the coverage of the complete biodiversity.*

*More information from reviewer #2: "Preserving and making reference material available also imposes a very strict quality control (QC) on the viability and the authenticity of living stock cultures/collections that are maintained/preserved. One must realize that this QC is crucial and that a correct application of the implemented QC system is key for the operational value of the whole system. Indeed, living stocks are subjected to genetic drifting during their preservation period. If the cultures are used for taxonomic references, the taxonomic markers are usually stable enough for identification above the strain/specimen level. However, when markers are used for identification at or even below the strain level (isolate/specimen level) the reference strains may have undergone genetic changes that make them no longer suited as references for identification, or in those cases where the living stocks are used in certified test in e.g. pharmaceutical context."*

To address the first part of the comment by the reviewer, we added a new short section entitled “Living Collections Capture an Important, Yet Minor Fraction of Extant Biodiversity”. To address the issue of quality control and collection integrity, we added a second short new section entitled “Challenges to Maintaining Collection Integrity”.

*3) Although the problem of integration of data is mentioned in the manuscript, it is not discussed deeply enough to underpin the lack of interest for the situation along with the lack of financial support.*

*More information from reviewer #2: "Researchers often exchange material in collections (mostly for free). In many cases, neither the donating nor the receiving party have the skills or the attitude to control the authenticity of the material that is exchanged resulting in the use of non-equivalent material for their research. This means that the results of cumulative research must be treated with a degree of criticism. Live stocks are also exchanged between collections and in this cases an important number of discrepancies of various nature are reported. If researchers plan to integrate data from different collections several controls need to be introduced as has been demonstrated by the various papers of the strainInfo team (Van Brabant B, Gray T, Verslyppe B, Kyrpides N, Dietrich K, Glockner FO, Cole J, Farris R, Schriml LM, De Vos P, De Baets B, Field D, Dawyndt P. 2008. Laying the foundation for a Genomic Rosetta Stone: creating information hubs through the use of consensus identifiers. IOMICS 12:123-127. Verslyppe, B., De Smet, W., De Baets, B., De Vos, P., Dawyndt, P. (2011). Make Histri: Reconstructing the exchange history of bacterial and archaeal type strains. Syst Appl Microbiol 34: 328-336. Verslyppe, B., De Smet, W., De Baets, B., De Vos, P., Dawyndt, P. (2014). StrainInfo introduces electronic passports for microorganisms. Syst. Appl. Microbiol. 37(1): 42-50 etc.)"*

The revised manuscript includes the following paragraph as part of the newly added “Challenges to Maintaining Collection Integrity” section: “Materials in collections are usually deposited by independent researchers and may be exchanged between stock centers, generating additional challenges in controlling the authenticity and equivalency of the stocks. […] To what extent a similar data integration and tracking system that follows each stock can be adopted by other communities is unclear, although a persistent uniform resource identifier would help deal with this issue.”

*4) Include more in-depth discussion of some of the ethical issues surround living collections:*

*Often the research has been supported by public money and hence the data belong to the public domain. However, protective actions are seldom undertaken to prevent this data being used for private, commercialized applications. Although the international community tried to take care of this problem via treatments and conventions (CBD, Nagoya protocol). Unfortunately, these conventions are not recognized by all governments on the globe and are more like a burden.*

*Only about 10-15% of the biological material that has been the subject of research is available via public collections. It is clear that from a scientific viewpoint, this is unacceptable and should be of major concern.*

We don't think that this is necessarily an issue that can be discussed constructively in this perspective for the following reasons: 1) Depending on the funding source and the community that each center serves, there is a wide range of ethical issues associated with the stocks that the centers in which the authors are involved distribute. 2) The perspective does address the issue of MTAs, and MTAs do address the issue of how stocks should be used. 3) The discussion of how public domain research his being used for profit by industry is one that goes way beyond the issues that this centers are involved with.